# Kea show three signatures of domain-general statistical inference

Amalia P.M. Bastos [1✉] & Alex H. Taylor [1]

One key aspect of domain-general thought is the ability to integrate information across different cognitive domains. Here, we tested whether kea (*Nestor notabilis*) can use relative quantities when predicting sampling outcomes, and then integrate both physical information about the presence of a barrier, and social information about the biased sampling of an experimenter, into their predictions. Our results show that kea exhibit three signatures of statistical inference, and therefore can integrate knowledge across different cognitive domains to flexibly adjust their predictions of sampling events. This result provides evidence that true statistical inference is found outside of the great apes, and that aspects of domain-general thinking can convergently evolve in brains with a highly different structure from primates. This has important implications not only for our understanding of how intelligence evolves, but also for research focused on how to create artificial domain-general thought processes.

[1] School of Psychology, The University of Auckland, Private Bag 92019, Auckland 1142, New Zealand. ✉email: a.bastos@auckland.ac.nz

There is currently great debate on the extent to which both human and nonhuman intelligence is domain specific[1–3] or domain general[4–7]: that is, whether subunits of the mind have evolved to solve specific adaptive problems, or whether intelligence evolves more generally, with the same cognitive mechanisms applied flexibly to multiple problems[8]. In humans, one source of evidence for domain-general intelligence, rather than domain-specific intelligence, are correlations between performance at different tasks ('g')[4,9]. Further evidence for domain generality in humans comes from our ability to transfer and combine information across different domains[7,10–12]. In animals, while there is some evidence for 'g'[13–16], this remains controversial[1,7,17–20], and there is currently little evidence for cross-modular integration of information[7,10,18,21]. This has led to claims that such integration is unique to humans[1,21–24] and dependent on language[10,11].

Reasoning under uncertainty is a central part of human decision-making[25,26]. Making inferences about uncertainty involves generating logical predictions about future events based on limited information[27]. This ability emerges much earlier in human development than expected for such an advanced form of cognition[28,29], and this type of reasoning has a number of key characteristics. First, when observing sampling events with a large number of objects, infants show true statistical inference, using the relative frequency of objects in a population to infer the most likely sampling outcome, rather than using quantity heuristics based on the absolute number of objects[28]. Second, infants can integrate information about physical constraints into their statistical inferences[30–32]. For example, infants override predictions based purely on relative probabilities when some objects in a population cannot be sampled because they are held back by a physical barrier[30]. Third, infants integrate social information about the preferences of a sampler into their statistical inferences, using their knowledge of an individual's bias to again override predictions based purely on relative probabilities. When an agent shows a preference by consistently selecting a minority item from a population, infants integrate this knowledge into their sampling predictions and expect biased sampling in the future[33–37]. These results suggest that infant statistical inference has three signatures: it uses relative frequencies (Signature 1) and is domain general, as infants can make predictions that integrate relative frequency judgements with information from both the physical domain (Signature 2) and the social domain (Signature 3).

Great apes are the only nonhuman species that have demonstrated true statistical inference, as they use the relative numbers of items within and between populations when predicting sampling events[38,39], rather than using quantity heuristics based on the absolute number of positive or negative objects. In contrast, capuchins use quantity heuristics based on the absolute frequency of negative items[40], and it is not yet clear whether rhesus monkeys, long-tailed macaques, pigeons and African grey parrots use relative frequency or the absolute number of either positive or negative items (or events) when predicting sampling outcomes[41–44]. At present, there is no evidence that any nonhuman animal can take physical constraints into account during sampling, but chimpanzees are capable of integrating social information about the preferences of a sampler into statistical inference. When given the choice of two experimenters who had previously both sampled preferred food items from a population, chimpanzees preferred to take a hidden sample from the experimenter who had shown a preference for picking the preferred food item from an unfavourable population[45].

Birds are an ideal group to test for domain-general statistical inference. This group has shown evidence not only of complex cognition[46,47], but also of behaviour suggestive of domain-general intelligence[48,49]. Here, we examined whether the kea, a parrot species endemic to New Zealand, show three signatures of human statistical inference, using comparable tasks to those administered to infants[28,30,31,33] and primates[38,40,45,50].

## Results

**Experiment 1**. In Experiment 1, we presented six kea with three tasks where they watched sampling events from two populations of mixed objects (rewarding and unrewarding tokens) at different proportions (illustrations of populations used are provided in Fig. 1). Over the course of three conditions, we tested whether kea could make sampling predictions using relative rather than absolute quantities. In Condition 1, subjects were presented with two populations, one with 100 rewarding tokens and 20 unrewarding tokens, the other with these frequencies reversed. Kea observed an experimenter sampling from each of these populations and were then presented with two closed hands. Three of six kea spontaneously showed a preference for the hand that had sampled the population with 100 rewarding tokens within their first 20 trials (Bayesian binomial test, relative preference 0.5, BF > 3, Table 1). We then gave all kea experience with this task until they chose the hand sampling the population with 100 rewarding tokens in 17/20 trials, which took kea 120 trials ($s = 61.97$ trials) on average (see individual learning data summaries in Supplementary Table 2).

Condition 2 tested if kea were using an absolute quantity-based heuristic by selecting the jar with the most rewarding tokens. Here, kea had to choose between 2 hands that both sampled from a population containing 20 rewarding tokens. However, one population had 100 unrewarding tokens, and the other had 4 unrewarding tokens. Four kea chose the jar with fewer unrewarding tokens within their first 20 trials (Bayesian binomial test, relative preference 0.5, BF > 3). Again, we gave subjects experience with this task until they reached a 17/20 criterion, which took 66.67 trials ($s = 41.31$ trials) on average.

Condition 3 controlled for a second absolute quantity-based heuristic, the avoidance of the jar with the most unrewarding tokens, a control which capuchins fail[40]. Kea observed sampling from one jar which contained 63 unrewarding tokens and 57 rewarding tokens and a second that again contained 63 unrewarding tokens but only 3 rewarding tokens. All 6 kea chose the jar with 57 rewarding tokens above chance within their first 20 trials (Bayesian binomial test, relative preference 0.5, BF > 3), and took 46.67 trials ($s = 27.33$ trials) on average to reach the 17/20 criterion.

The results of Experiment 1 (Table 1) provide conclusive evidence that kea show true statistical inference using the relative frequency of items, rather than using quantity heuristics based on the absolute number of items. Four of our six subjects performed above chance within their first 20 trials of both Conditions 2 and 3, indicating they had not learnt during their past experience to use a heuristic based on choosing the population with either the most rewarding, or most unrewarding items. Kea therefore did not behave as capuchins do, in using the absolute number of positive or negative tokens within the jars to make decisions[40,50]. Instead, kea mirrored the performance of infants[28] and chimpanzees[38] in using the relative frequency of objects across this experiment.

**Experiment 2**. Experiment 2 tested whether kea could integrate information about a physical constraint into their prediction of a sampling event, as infants do[30]. After giving kea experience of a barrier (training protocols provided in Supplementary Methods), we presented them with two tasks where two jars, each with a barrier placed half-way down the jar, were sampled from. Each jar contained identical overall populations of tokens (80 tokens total:

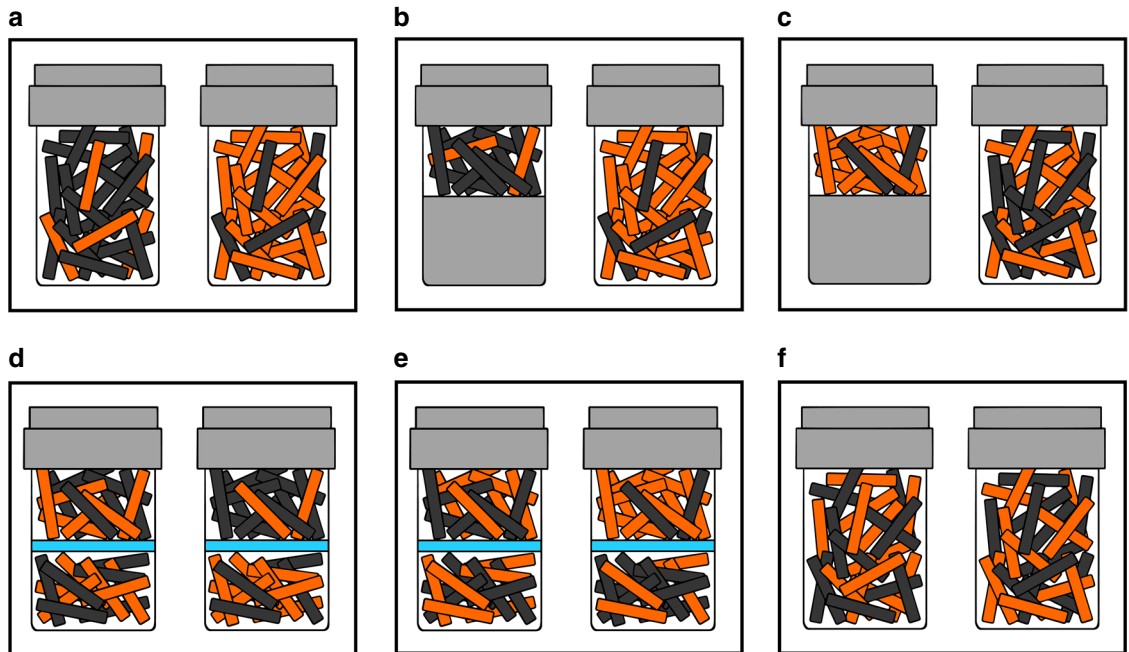

**Fig. 1 Token populations used across all experimental conditions.** Proportional representation of token populations for Experiments 1–3. In the illustrations, orange rectangles represent the unrewarding tokens, and black rectangles represent the rewarding tokens. **a–c** The token frequencies for Condition 1, Condition 2 and Condition 3. **d**, **e** The token frequencies for Experiment 2, with the blue lines representing a physical barrier. **f** The token frequencies at test for Experiment 3.

| **Table 1 Individual performance in experiments 1–3.** | | | | | |
|---|---|---|---|---|---|
| | **Experiment 1** | **Experiment 1** | **Experiment 1** | **Experiment 2** | **Experiment 2** | **Experiment 3** |
| | **Condition 1** | **Condition 2** | **Condition 3** | **Condition 1** | **Condition 2** | **Condition 1** |
| Blofeld | 10/20 (BF = 0.27) | 14/20 (BF = 1.29) | **15/20 (BF = 3.22)** | **18/20 (BF = 262.80)** | **15/20 (BF = 3.22)** | 12/20 (BF = 0.40) |
| Bruce | 12/20 (BF = 0.40) | **16/20 (BF = 10.31)** | **16/20 (BF = 10.31)** | **17/20 (BF = 43.80)** | **15/20 (BF = 3.22)** | **17/20 (BF = 43.80)** |
| Loki | **15/20 (BF = 3.22)** | **16/20 (BF = 10.31)** | **19/20 (BF = 2496.61)** | **18/20 (BF = 262.80)** | **19/20 (BF = 2496.61)** | 11/20 (BF = 0.30) |
| Neo | 14/20 (BF = 1.29) | 14/20 (BF = 1.29) | **15/20 (BF = 3.22)** | **15/20 (BF = 3.22)** | 14/20 (BF = 1.29) | **15/20 (BF = 3.22)** |
| Plankton | **15/20 (BF = 3.22)** | **15/20 (BF = 3.22)** | **16/20 (BF = 10.31)** | **17/20 (BF = 43.80)** | **17/20 (BF = 43.80)** | 10/20 (BF = 0.27) |
| Taz | **15/20 (BF = 3.22)** | **16/20 (BF = 10.31)** | **17/20 (BF = 43.80)** | **17/20 (BF = 43.80)** | **16/20 (BF = 10.31)** | **17/20 (BF = 43.80)** |

Number of correct trials performed by each subject ($n = 6$) within the first block of 20 trials for each condition of the three Experiments. In Experiment 3, correct trials constituted trials in which the subject chose the biased sampler, E1. Which of the two experimenters was the biased sampler (E1) was counterbalanced across subjects split into two groups: Neo, Bruce, Blofeld; and Loki, Plankton, Taz. Performance was tested using two-tailed Bayesian binomial tests (test value of 0.5, default Beta prior parameters at 1.0). Values with a Bayes Factor greater than 3 are shown in bold.

40 rewarding and 40 unrewarding) but the proportions differed above and below the barriers. In Condition 1, one jar contained 20 rewarding and 20 unrewarding tokens above the barrier, and the same below it. The other contained 20 rewarding and 4 unrewarding tokens above the barrier and the remaining 56 tokens below the barrier (20 rewarding, 36 unrewarding). Condition 2 was identical but with the frequencies reversed. In one jar, one population contained 20 rewarding tokens and 20 unrewarding tokens above and below the barrier. The other contained 4 rewarding, and 20 unrewarding tokens above the barrier, and the remaining tokens below it (36 rewarding, 20 unrewarding). If kea could integrate knowledge of the barrier into their relative frequency judgements, we predicted they would choose the jar with 20 rewarding tokens and 4 unrewarding tokens above the barrier in Condition 1 and the jar with equal numbers of rewarding and non-rewarding tokens above the barrier in Condition 2.

As in Experiment 1, kea were given further training until they reached a criterion of 17/20 trials, which took them, on average,

26.67 trials ($s = 16.33$ trials) for Condition 1 and 48 trials ($s = 30.33$ trials) for Condition 2. Summaries of individual training data are provided in Supplementary Table 3. Five of the six subjects tested performed above chance in the first 20 trials of both conditions of Experiment 2 (Bayesian binomial test, relative preference 0.5, BF > 3, Table 1).

These results not only confirm the results of Experiment 1, in showing kea use the relative frequency of objects to make statistical inference, but also show that kea can flexibly integrate physical knowledge into these inferences. When a barrier was placed in the jar, kea used only the relative frequency above the barrier when deciding which hand was more likely to contain a rewarding token.

**Experiment 3**. Experiment 3 investigated whether kea could integrate social information about sampler biases into their predictions. We closely matched the procedure used in chimpanzees[45], providing kea with experience of a biased and an unbiased sampler. We first tested whether kea could distinguish between

two human experimenters. Kea observed one experimenter closing their hand over a token and another experimenter who held nothing in their closed fist. The experimenters then either switched position or stayed in the same place and the kea was given the choice of one of the two experimenters' hands (training procedures detailed in Supplementary Methods). All kea passed this task above chance within 1 session of 20 trials. We then tested kea for pre-existing preferences for one of the two experimenters in a token-exchange task, where both experimenters requested a rewarding token for exchange simultaneously and kea could select which of the two experimenters to obtain a reward from. We continued with this training until kea selected one of the experiments at between 9/20 and 11/20, which they all did within three sessions.

Kea then observed demonstrations of biased and unbiased sampling from the two experimenters. While the biased sampler selected rewarding tokens from a population of 110 tokens containing 10 rewarding and 100 unrewarding tokens, the unbiased sampler selected rewarding tokens from a population of 10 unrewarding and 100 rewarding tokens. Therefore, during demonstrations, both samplers were equally associated with a rewarding sampling outcome: both experimenters always sampled a rewarding token, but whilst the biased sampler did so by looking into a population with a minority of rewarding tokens, the unbiased sampler did so by blindly sampling from a population with a majority of rewarding tokens.

At test, kea observed as the same two samplers picked from populations with an equal number of rewarding and unrewarding tokens (55 rewarding and 55 unrewarding tokens). If kea continued to use the relative frequencies of the tokens in each jar, we expected them to choose at chance. In contrast, if the kea understood that the biased sampler was indeed biased to choose a rewarding token, while the unbiased sampler had only been choosing rewarding tokens at the same frequency as the biased sampler due to the populations they were sampling from, kea should choose the biased sampler at test. This was because while the unbiased sampler would now be likely to choose a rewarding token half the time, the biased sampler should continue to choose the rewarding token in every trial. In order to succeed at this task, kea would need to integrate the social knowledge acquired during the demonstration phase into their probabilistic sampling predictions. Three of the six kea chose the biased sampler above chance at test (Table 1). None of these three kea had previously shown a preference for either experimenter in the token-exchange task or during the demonstration phase (see Supplementary Table 4). These results therefore mirror those from infants[33] and chimpanzees[45] in showing that kea can integrate social information on sampler biases into their statistical inferences.

**First-trial performances**. In order to ensure that kea's performance was not merely a consequence of associative learning within the first 20 trials of each condition, we analysed first-trial performance across the three experiments. Across all conditions and all experiments, subjects' first trials were correct in 72.22% of trials. Taking into account only the subjects that succeeded within the first 20 trials of each condition, first-trial performances were correct in 81.48% of cases. We fitted an intercept-only Bayesian model to our first-trial data for all subjects. When compared against a 0.5 baseline probability of success, our model revealed that the median posterior probability of a randomly selected kea succeeding within their first trial, regardless of condition, was 0.70 (pMCMC = 0.005).

**Learning effect analyses**. We ran Bayesian correlation tests of average performance across the first 20 trials of each condition to

examine whether performance increased over the course of the first 20 trials. We found no evidence for learning effects (BF < 3; results for each condition's analysis are reported in Supplementary Table 5). Similarly, we examined if performance increased across conditions, but found no correlation between condition number and average performance within the first 20 trials of each condition (BF = 0.508). These results show that subjects' performance did not improve over time within or between test conditions, and therefore kea were not relying on low-level associative strategies to make their sampling predictions.

## Discussion

Our study shows that kea display three signatures of human statistical inference. Experiment 1 shows that, like infants[28,30,31,33] and chimpanzees[38,45], kea showed evidence of true statistical inference. Kea predicted likely sampling outcomes based on the ratio of objects in the populations being sampled from, rather than using quantity heuristics, such as selecting the population with the greatest number of positive tokens, or avoiding the population with the most negative tokens. Experiment 2 shows that kea, like infants[30–32], can integrate knowledge about a physical barrier into their predictions of a sampling outcome, even though the overall population distributions were identical. Finally, Experiment 3 shows that, like infants[33] and chimpanzees[45], kea are capable of integrating social information about biased and unbiased samplers into their predictions. As in past work on chimpanzees[45], kea took observed information about the biases of experimenters and integrated it into their predictions of what would occur when these experimenters sampled from equally distributed populations of objects. Therefore, just like infants and the great apes, kea made statistical inferences using relative rather than absolute quantities, and then integrated social and physical information into their predictions, using their knowledge of physical barriers and the bias of a sampler, to override predictions based purely on relative probabilities[31].

To observe these three signatures of domain-general statistical inference in kea is surprising, even given recent developments in avian cognition. Much work in this field over the past 15 years has focused on corvids, which, as a group, have produced their most impressive problem-solving performances predominantly on domain-specific, ecologically relevant tasks, such as those involving caching or tool use[51–60]. Parrots have only recently become the focus of a sustained, global research effort[61]. Behaviour suggestive of more domain-general processes have emerged in both groups from studies examining the ability of non-tool users to solve tool problems[48,49,62–66]. Our results both support these claims and greatly extend them, in showing that the integration of very different types of information—concerning physical barriers and sampling biases—into statistical inferences is possible in at least the mind of one parrot species. Furthermore, both the token transfer behaviours, and the observation of sampling events from a population of objects, had very little ecological relevance to the kea, a species that feeds by extractive foraging in alpine and sub-alpine environments[67]. Despite this, kea not only learned to solve these problems, but did so by using domain-general statistical inference, rather than quantity heuristics.

It is important to note, however, that the level of information integration across our experiments was different. Experiment 2 tested if kea could simultaneously integrate information about relative frequency with the presence of a barrier in order to make a judgement, because relying solely on one source of information—either the presence of a barrier, or the overall frequency of the population—would have led to the incorrect choice.

In Experiment 3, however, kea needed to override their prior reliance on relative frequency information in favour of information about the sampling bias of an experimenter. Our results therefore suggest that kea can fully integrate physical information with relative frequency by using both sources of information simultaneously, and can integrate social information by over-riding relative frequency information with social information. However, more work is needed to show if kea can also simultaneously consider social and relative frequency information, or can make judgements that simultaneously combine social, physical and relative frequency information together.

Unlike previous studies in primates[38–40,44,45,50], kea were presented with additional training trials before progressing to the next condition. This procedure is commonly used during cognitive experiments on birds[68–70] to ensure subjects are consistent at a specific condition before being given a novel one. It seems unlikely this additional training would have affected the strategy kea were using to make their sampling predictions across our experiments, as this additional experience would have only consolidated whatever strategy kea were using at the time, be it the use of relative frequencies or quantity heuristics. That is, the learning of the kea was unstructured: both quantity heuristics and domain-general statistical inference would have worked to solve several of the early problems presented to kea, yet kea clearly mirrored the type of statistical inferences made by humans and the great apes when solving these problems.

It is currently unclear how infants, apes and the kea in this study extract statistical information. In particular, as Denison and Xu[28] note, it is not yet clear if subjects make inferences using discrete or continuous quantity representations, but this does not detract from the results here; either way, subjects were extracting information about the relative frequencies of objects, using it to make predictions about the relative probability of reward (Experiment 1), and then integrating these judgements with both physical (Experiment 2) and social (Experiment 3) information. One key area for future work will be determining exactly how statistical information is extracted and represented by humans, apes and kea.

Birds last shared a common ancestor with humans at least 312 million years ago[71]. This evolutionary distance suggests that domain-general statistical inference may have arisen twice on our planet via convergent evolution. However, further work is required over a wider range of avian and primate species to provide a more accurate evolutionary account of when statistical inference has emerged in different taxa, and so test this hypothesis further.

The statistical inference abilities observed in our study are particularly interesting given that birds have brains with a much smaller absolute size than humans, a very different structure[47], and much higher neuronal densities[72]. Our results therefore suggest that (i) aspects of domain general thought can evolve multiple times, rather than being a one off or a product of a highly unlikely sequence of evolutionary events and (ii) that a particular brain architecture, specifically a layered cortex, is not necessary for this type of higher-order intelligence to evolve. This has important implications not only for our understanding of how intelligence evolves, but also for research focused on creating artificial, domain general thought processes (artificial general intelligence), specifically on the degree to which such processes should mirror mammalian cortical processes and structures[73–76].

## Methods

**Subjects and apparatus**. Our subjects were six male kea at Willowbank Wildlife Reserve (see Supplementary Table 1). Kea were housed in a large outdoor aviary, where food and water were available ad libitum. Participation in the study was voluntary and subjects were free to leave mid-session at any point. This research

was conducted under ethics approval from The University of Auckland Ethics Committee (reference number 001816).

Each subject was allocated an individual training platform (42 cm × 42 cm) within the aviary on which they were tested. Performance in trials was rewarded with soaked Science Hill Diet pellets. A small wooden shelf (60 cm × 20 cm) with a plexiglass screen (43 cm × 29 cm) was used to separate subjects from the apparatus and the experimenter during testing. Transparent jars (∅10.5 cm, 16 cm tall) were used during training and testing which contained populations of either rewarding (black) or unrewarding (orange) wooden tokens (7 cm × 1 cm × 1 cm). Each jar held a maximum of 120 tokens. When the jars were too large for a population of tokens, tokens sat on a cardboard platform that was placed inside the jar, to ensure subjects could not see the experimenter's hands during sampling. Semicircular cardboard lids (∅11.5 cm, 5.5 cm tall) were attached to the top of each jar to ensure subjects could not see which tokens were being sampled. Where barriers were used, a blue foam disk (∅10.5 cm, 1 cm thick) was added into the jar.

**Training and procedures**. Throughout training and testing, subjects were required to select which of two closed hands contained an out-of-sight rewarding token, while ignoring the hand containing the unrewarding token. The rewarding token would then be exchanged with the experimenter for a food reward. Where subjects attempted to exchange an unrewarding token, this was taken by the experimenter but not rewarded.

Subjects were trained to attend to and track hand trajectories for a previous study. Subjects were trained specifically for this study on hand tracking so they could follow the motion of sampling, and making inferences about sampling from token populations in two jars, by selecting a hand that picked a rewarding token from a population of 100% rewarding tokens, over a hand that sampled from a population of 100% unrewarding tokens (see Supplementary Methods). In order to allow for a full counterbalancing of trial presentations at test and minimise side biasing, subjects were also taught to simultaneously attend to the side on which jars were placed and whether hands were presented in parallel or crossed over. This was trained over four separate training steps (outlined in detail in the Supplementary Methods).

Before each experimental session, subjects were given motivation trials, where they had to select and exchange a rewarding (black) token and ignore a nearby unrewarding (orange) token with the experimenter three times in a row, prior to the start of the session. This ensured subjects were eager to work and remembered which of the two tokens they should search for at test. Testing was carried out by three experimenters who were blind to experimental design and hypotheses, wearing mirrored sunglasses. Subjects only proceeded to the next testing condition or experiment upon reaching a criterion of 17/20 correct choices within the same block, or completing 240 trials (12 blocks) without reaching criterion. This ensured that subjects were confident in the current task before proceeding to a more demanding one. Throughout testing, hand presentation (parallel or crossed), and location of the rewarding hand at time of choice were all pseudorandomised and counterbalanced within blocks of 20 trials. Throughout training and testing, kea could only see the experimenter's hand disappear behind the cardboard occluder on the top of the jar. Therefore, subjects were unable to see how far down the populations the experimenter's hand reached, or which token it sampled from the population. In test conditions with very disparate ratios of rewarding-to-unrewarding tokens, we ensured that at least two tokens from the minority population were fully visible to the subjects in every trial.

**Experiment 1**. This experiment investigated whether kea are able to make statistical inferences about two populations of objects using relative frequencies. Over three conditions, we tested whether (1) kea would prefer a sample from a population containing a majority of rewarding tokens, as opposed to a population where they were in the minority, and whether kea rely on relative frequencies, (2) the absolute number of rewarding tokens or (3) the absolute number of unrewarding tokens, when choosing between samples from two populations. Illustrations for the three conditions are provided in Fig. 1.

The first condition aimed to test whether kea would prefer a sampled token from a population where there was a higher probability of randomly sampling a rewarding token, as opposed to a population where there was a higher probability of sampling an unrewarding token. Two jars were presented: one contained a 1:5 ratio of rewarding-to-unrewarding (rewarding-to-unrewarding tokens), and the other contained a 5:1 rewarding-to-unrewarding ratio. Both jars contained 120 tokens in total.

The second condition tested whether kea were making their choices based on absolute frequencies or relative frequencies. In order to make this distinction, subjects were presented with two jars containing the same number of rewarding (black) tokens, in differing proportions. One jar had a 1:5 rewarding-to-unrewarding population of 120 tokens, whilst the other had a 5:1 rewarding-to-unrewarding population of 24 tokens. If kea were using the absolute number of rewarding tokens to guide their choices, we predicted they would choose at chance. If, in contrast, they were taking into account the relative proportion of rewarding-to-unrewarding tokens, we predicted they would choose the jar with only four unrewarding tokens.

In the third condition, we presented subjects with two jars containing the same number of unrewarding tokens: one jar had a 57:63 rewarding-to-unrewarding

population (120 tokens total), whilst the other had a 3:63 rewarding-to-unrewarding population (66 tokens total). If kea were simply selecting the jar containing the fewest unrewarding tokens rather than comparing between the frequencies of token populations between jars, they should perform at chance in this condition.

**Experiment 2**. This experiment investigated whether kea are able to integrate physical constraints into their sampling inferences. We first gave kea both ego-centric and allocentric experience of a foam barrier. Kea were first allowed to sample from two small jars (ø6 cm, 7.5 cm tall) where a population of 20 rewarding tokens was either physically accessible or impeded by a barrier, then observed as an experimenter attempted to sample from two populations with a similar config-uration (details of training are provided in the Supplementary Methods). Over two conditions, we presented kea with two populations of tokens which were split in the middle by physical barriers, and tested whether kea understood that only the population above the barrier could be sampled from. Both conditions are illustrated in Supplementary Table 1.

In the first condition, both jars each contained 40 rewarding and 40 unrewarding tokens. One jar had a 1:1 rewarding-to-unrewarding population (40 tokens) both above and below the barrier, and the other had a 5:1 rewarding-to-unrewarding population (24 tokens) above the barrier and 5:9 rewarding-to-unrewarding population (56 tokens) below it. This was used to test whether kea were simply attending to which jar had the largest number of rewarding tokens near the top, which should lead to performance at chance, as opposed to comparing between the relative frequencies of tokens for the two accessible populations. Subjects were also expected to perform at chance in this condition if they were comparing between the token frequencies of the two jars without taking the barrier into account, as both jars contained the same absolute number and relative frequencies of rewarding and unrewarding tokens, 1:1 (40 rewarding, 40 unrewarding).

The second condition was identical, but with reversed proportions: one jar had a 1:1 rewarding-to-unrewarding population of 40 tokens above and below the barrier, whilst the other had a 1:5 rewarding-to-unrewarding population (24) tokens above the barrier and the remaining 9:5 rewarding-to-unrewarding population below it. This condition tested whether kea were selecting the jar with the fewest unrewarding tokens near the top, in which case they should perform at chance, or comparing between the relative frequencies of the two accessible populations in the two jars. Again, both jars contained the same absolute number and relative frequencies of rewarding and unrewarding tokens.

**Experiment 3**. Experiment 3 tested whether kea could take a biased sampler's biases into account during a sampling event. Two experimenters were randomly assigned and counterbalanced between birds as either unbiased (hereafter 'E2') or biased (hereafter 'E1'). The procedure was based on the study by Eckert and colleagues[22] with chimpanzees, and required four experience phases.

In the first phase, we ensured that kea could tell the difference between the two experimenters: E1 and E2 stood next to each other and either picked up a food pellet or nothing into their right hand, then closed their fist. E1 and E2 either switched sides or stood on the same side for 5 s, before calling the subject's name in turn and presenting their hands simultaneously for the subject to make a choice. The experimenter's sides, the order of their actions, whether or not they switched sides (and whether the experimenter that switched sides did so by walking behind or in front of the other), and the order in which the subject's name was called, were all pseudorandomised and counterbalanced within sessions of ten trials. Subjects received this training until they achieved a 17/20 criterion.

Following this, subjects were given a preference test. E1 and E2 offered an empty hand to the subject as it held a rewarding token. The subject then had a choice of whom to deliver the token to, in exchange for a reward. Which experimenter placed the token on the platform and the side on which each experimenter stood were pseudorandomised and counterbalanced within blocks of 20 trials. In order to proceed to the next stage, subjects were required to show no preference for either experimenter, that is, select E1 at between 9/20 and 11/20.

Subjects then observed demonstrations by the two experimenters where they had the opportunity to learn that E2 picked randomly from tokens, whilst E1 acted as a biased sampler. For the demonstration, E1 and E2 stood next to each other and neither wore mirrored sunglasses so the kea could see their eyes. E2 always had a 10:1 rewarding-to-unrewarding population of 110 tokens, whilst E1 always had a 1:10 rewarding-to-unrewarding population of 110 tokens. Therefore, based on sampling probability alone, E2 was far more likely to sample a rewarding token than E1. During the demonstrations, E1 and E2 took turns sampling, and E2 always tilted their heads back and looked up whilst sampling, whilst E1 lowered their heads close to the jar and looked into it as they made a choice, keeping their hands in the jar for 3 s. Both experimenters always sampled a rewarding token, so that they were equally reinforced. After sampling, either both experimenters stood on the same side for 5 s, or switched sides, before presenting their closed fists to the subject simultaneously. Which side each experimenter stood on, who sampled first, whether or not they switched sides (and whether they did so by going behind or in front of the other experimenter), were all pseudorandomised and counterbalanced within sessions of ten trials. In order to proceed to the next experience phase, subjects had to select E1 at 9/20 or above, showing that they had no preference for E2 and were therefore not simply attending to the token populations within jars

during demonstrations. All subjects passed this criterion within 20 trials except for Neo, who experienced two blocks (40 trials) of demonstrations.

The final experience phase before test was a memory probe. In this phase, E2 presented each bird with a block of 20 trials where 2 jars of 120 tokens each contained either 100% rewarding or 100% unrewarding tokens. E2 wore mirrored sunglasses for this phase, and presented their hands in parallel or crossed over, as in previous experiments. This was done by E2 because they were the unbiased experimenter. We predicted that if greater exposure to one or another person before test could affect test results, then carrying out an extra set of trials with E2 would make the choice of E1 less likely at test. Similarly, an increased number of positive 'rewarding token' experiences with E2 should make the choice of E1 less likely at test. Jar order and hand presentation were counterbalanced and pseudorandomised. This phase ensured that subjects could and would still attend to the contents of jars following the demonstrations, and had not simply learned to ignore jar contents during the demonstration phase.

Subjects were then given the experimental task. They observed three trials of demonstrations identical to before, and then jars were swapped to 1:1 rewarding-to-unrewarding populations of 110 tokens. Based on token probability alone, E1 and E2 were now equally likely to sample a rewarding token. However, E1 and E2 behaved in identical fashion to demonstration trials, suggesting that they were biased and unbiased samplers, respectively. At test, E2 sampled truly randomly, whilst E1 continued to sample only the rewarding token in each trial. We expected that if kea understood that E1 was a biased sampler, they should choose them significantly above chance.

**Analyses**. All trials were filmed and coded in situ. Subject performance was blind coded for 10% of all video data and compared to in situ coded data. Inter-observer reliability was high (Cohen's kappa = 1.0). Performance in the first 20 trials of each condition were analysed at the individual level, using two-tailed Bayesian binomial tests with a test value of 0.5. We used Bayesian correlation tests to investigate average performance across the first 20 trials of each condition over trial number, and average performance on the first 20 trials of each condition across the 6 experimental conditions. We used default parameters (non-directional correlation, prior width = 1) for all correlation tests. These statistical analyses were carried out in JASP 0.9.2[77]. We followed the convention that a Bayes factor (BF) < 0.33 shows substantial support for the null hypothesis, whilst a BF > 3 shows substantial support for the competing hypothesis[78].

We also analysed first-trial performance at the group level using a Bayesian intercept-only model, using a Bernoulli distribution. We fitted our model to all thirty-six first-trial data points, across all individuals and conditions. Intercepts were given weakly informative Gaussian priors (M = 0, SD = 1), to reduce overfitting. Reported pMCMC values reflect the probability of performance differing from a 0.5 chance baseline. This analysis was conducted in R 3.4.1[79] using the "brms" package[80]. We used Stan to run Hamiltonian Monte Carlo estimations[81].

All raw data is available in Supplementary Data 1. Code and MCMC chain diagnostics are also provided as Supplementary Information.

**Reporting summary**. Further information on research design is available in the Nature Research Reporting Summary linked to this article.

## Data availability
Our full dataset is available in Supplementary Data 1.

## Code availability
Our code, MCMC chain diagnostics, and prior distribution details are available in the Supplementary Information file.

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

## Acknowledgements

We thank Rachel Johnston, Rosalie Molin and Amey Zhang for their help with data collection. We also thank Amey Zhang for the illustrations of our apparatus, Zoe Tai for blind-coding our video data and Scott Claessens for his help with our analyses. We are also grateful to Willowbank Wildlife Reserve staff, particularly Nick Ackroyd, for their help with our research. This project was made possible through the support of a grant from Templeton World Charity Foundation (AHT), a Rutherford Discovery Fellowship (AHT) and a Prime Minister's McDiarmid Emerging Scientist Prize (AHT).

## Author contributions

A.P.M.B. and A.H.T. designed the experiments; A.P.M.B. and three experimenters blind to hypotheses carried out training and data collection; A.P.M.B. analysed the data; A.P.M.B. and A.H.T. wrote the paper. All authors gave approval for publication.

## Competing interests

The authors declare no competing interests.
