## [Peer Review File · Nature Communications]

Reviewers' Comments:

Reviewer #1:

Remarks to the Author:

The present study presented six kea individuals with tasks requiring statistical inferences from population to sample, using a similar methodology as has been applied in previous studies with human infants, nonhuman great apes and monkeys. In a series of three experiments, subjects seemingly not only used information about the proportional composition of populations in order to predict sampling outcomes (Experiment 1), but also seemed to consider information about physical (Experiment 2) and social constraints (Experiment 3). The authors conclude that kea are capable of true statistical inference- a capacity that has previously been found in only few nonhuman species (mainly primates). Moreover, the authors claim that the birds' capacities even exceed those of New and Old world monkeys: While capuchin monkeys and long-tailed macaques failed crucial test conditions controlling for the use of absolute quantity heuristics in previous studies, kea seemingly succeeded in similar controls in the present study. Lastly, a main conclusion of the paper is that kea can integrate information from different cognitive domains (physical and social) into their statistical inference and are therefore capable of domain-general thought. This conclusion leads the authors to speculate that domain-general statistical inference may have evolved in a convergent manner in distantly related species.

The topic of the present study is highly relevant, up to date, and interesting for a broad readership. So far, the study of intuitive statistics has mainly focused on (human and nonhuman) primates. However, to explore the evolutionary roots of this fundamental capacity, it is important to consider the possibility of convergent evolution and investigate this capacity in a wider range of species. Birds with complex cognitive abilities, such as parrots and corvids, seem promising candidates for finding statistical abilities outside the primate lineage. The present study, therefore, fills an important gap in the literature and will most likely stimulate further research in this area. The paper is well written, clear and concise. I particularly appreciate the authors' effort to test not only for the "standard" capacity of reasoning from population to sample, but also to investigate whether birds can use information from different cognitive domains when drawing statistical inference. Hence, I believe that this is a very promising, comprehensive, and relevant study, and the authors have the data to support most of their claims. That being said, I have several points of criticism/suggestions for improvement which I feel should be addressed by the authors.

Introduction: I think the introduction (especially the first paragraph) would benefit from more details. Some terms and concepts are not sufficiently explained for the readership of a multidisciplinary journal such as Nature communications. For instance, the authors never define what they mean with the term "cognitive domain", and many readers may not be familiar with "g". Considering the importance of these terms for the present study, I believe it would be wise to add more explanation in the beginning of the manuscript. The literature review seems fairly up to date. However, I would suggest adding the following important references:

Page 3, line 37: Teglas et al., 2011 (one of the most convincing demonstrations of infants integrating physical (spatio-temporal) and statistical information).

Page 4, line 48: Eckert, Call, et al., 2018 (the first study to rule out absolute quantity heuristics both regarding preferred and non-preferred food items in great apes);
Placi et al., 2019 (long-tailed macaques use relative, rather than absolute frequencies in a statistical reasoning task with a different task format than in previous studies).

Page 4, line 51: Roberts et al., 2018 (Pigeons use relative, rather than absolute quantities in a probabilistic reasoning task).

Both Roberts et al. (2018) and Placi et al. (2019) suggest that not only great apes may be capable of true statistical inference -> the authors might want to rephrase the sentence on page 4, lines 47f.

Experiment 1: My main criticism is that this study did not test a spontaneous and intuitive ability, but rather trained subjects to reach criteria in the different tasks, which is a crucial difference to previous studies on human infants, great apes, and monkeys. Hence, we do not know whether birds truly inferred the outcome of the sampling event, like primates did, but rather learned an association between one of the populations and a reward over the course of trials. In contrast to ape and monkey studies, the authors do not report first trial performance or test for an effect of trial number on the subjects' performance in order to shed more light on this possibility. This problem is aggravated by the fact that the authors did not counterbalance the order of conditions- hence, learning across conditions is possible, and perhaps even likely, especially considering that some of the populations remained the same across several conditions. For instance, both in Condition 1 and 2, the unfavorable population contained a 20:100 mixture. After having received on average 120 trials in Condition 1, kea most likely knew that the 20:100 population would not lead to a rewarding token as a sample when it was presented to them again in Condition 2- they did not need to draw any statistical inference. This learning explanation is supported by the fact that birds became gradually better over the course of the three test conditions, a fact that would otherwise be surprising considering that task difficulty increased and the magnitude of difference between the two populations decreased (which would make statistical reasoning presumably more difficult). I understand that for birds the general procedure might be less intuitive/familiar than for primates, and therefore more training might be necessary. However, ideally this training should have been administered as a pre-training only, and not involve the same populations as in the actual test conditions.

The authors mainly rely on the birds' good results in Condition 2 and 3 when claiming that kea, in contrast to monkeys, are capable of true statistical inference based on relative rather than absolute quantity. However, given the methodological discrepancies between the present study (several hundreds of trials per condition, using the same population across conditions) and primate studies (12 trials per condition, different populations for each condition), I think it is misleading to conclude that kea, but not monkeys, succeeded in statistical inference tasks in which absolute and relative frequencies were disentangled.

I suggest that the authors re-run Condition 2 with new populations (different ratios) to verify their claim. If this is not possible, the discussion should acknowledge the methodological differences between bird and monkey studies, as well as the alternative explanation that birds may have formed associations between populations and rewards in Experiment 1.

Experiment 2: For this experiment I have a question regarding the procedure, in particular what exactly the birds could see: Did they see that the experimenter's hand reached only into the upper compartment of the jar? Or did they just see that the hand was inserted into the jar, but not into which compartment it reached? I think this question is crucial, because it may tell us how the birds represented the scene. If kea saw that the experimenter's hand only reached the upper compartment, they did not necessarily have to consider both statistical and physical circumstances in order to solve the task; it would have been sufficient to perceive the two compartments as two separate populations- similar as if they would have been in two different jars. Again, this is an important difference to the infant studies testing for the integration of statistical and physical

information (e.g. Teglas et al., 2007; 2011; Denison & Xu 2010; Denison et al., 2014). In these studies, infants inferred the effect of the physical determinant (e.g. barrier) on the sampling event without actually seeing the effect (because the sampling event itself was occluded).

If, in the present study, kea did see the effect of the barrier (i.e. that the hand was only inserted in the upper container), the authors should acknowledge this difference to infant studies in their discussion and tune down the conclusions drawn from this experiment. Kea's performance in this experiment is still impressive and a first hint that birds might consider a physical constraint when drawing statistical inference. But the here demonstrated abilities are clearly qualitatively different from those that have been demonstrated in human infants.

Experiment 3: In this experiment I do not understand the rationale of the final experience phase (memory probe) with E2. Why was it beneficial to give kea extra experience with only one of the two experimenters before the actual test?

The beauty of the original experiment (which was developed for chimpanzees; Eckert, Rakoczy, et al. 2018) was that subjects were non-differentially rewarded by the two experimenters in the demonstration phase, before they were allowed to make a choice between them in the test phase. Hence, chimpanzees had no way of associating one of the two experimenters positively or negatively, and instead had to use their statistically acquired knowledge about the experimenters' biases as only source of information.

By contrast, in the present study, the authors included an additional experience phase in between the demonstration phase and the test phase. Crucially, in this phase the birds were given additional experience with only one of the two experimenters (E2), therefore weakening the non-differential character of the demonstration phase. In other words: The latest experience that birds had with each of the experimenters by the time the test started was "E1 always samples rewarding tokens", "E2 sometimes samples rewarding tokens and sometimes unrewarding tokens". Hence, it is unclear whether birds succeeded in the test condition truly by the means of statistically acquired knowledge or rather by a strategy to avoid the experimenter which had given them unrewarding samples in their previous encounter. Again, if the authors are not able to rule out this alternative explanation, they should acknowledge it in the discussion and tune down their interpretations.

Additional remarks: In my opinion, the term integration of information from different cognitive domains is not appropriate. What the authors could show in this study (if the above-mentioned points are sufficiently addressed/clarified) is that kea flexibly decided which of two sources of information (e.g. proportional distribution of items vs. information about a physical barrier) will determine the outcome of a sampling event. An integration of these two sources of information would imply that the influence of each factor is weighted and included in the final computation, leading to a gradual response (similar as in, e.g. Teglas et al., 2011).

The reference list contains some formatting errors.

Reviewer #2:

Remarks to the Author:

Bastos & Taylor

Kea (*Nestor notabilis*) show three signatures of domain-general statistical inference

This paper aims at testing whether members of a parrot species (Kea) show "true statistical inference". The general approach is to show the subjects transparent jars filled with various numbers of black and orange plastic items, which the birds are trained to exchange for food items, hence they are 'tokens'.

The paper presents 3 experiments with different conditions each, including keeping the number of +ve items constant but varying the number of -ve ones, and varying the absolute numbers as opposed to (or as well as) the proportions.

I enjoyed the paper and the experiments, but had one conceptual difficulty. All the conditions and conclusions are presented and discussed using frequentist descriptions (for instance, in one condition a jar contained 63 +ve and 57 -ve tokens while the other jar the reverse numbers). In fact, it is not possible for any observer (even a human) to 'count' the tokens in a jar, since they are distributed randomly in a volume so that some are behind. Further, it is unreasonable to suggest without evidence that the birds count such high numbers.

It is more likely that they use the 'blackness' and 'orangeness' that they can see to categorize the jars. That would be a useful heuristic for the probability of extracting a black or orange bar. In that case, the conditions can be described in terms of relative (or absolute) areas of a given colour. For instance, a jar with 20 +ve and 100 -ve tokens should be described as averaging 16.7% black area. This heuristic is perfectly correlated with changes in both absolute and relative numbers, and leads to using a relative assessment of probability of reward.

I am not clear on whether this change in the description would harm the interpretations the authors put forward, but would like to see that discussed. With frequencies, 2/10 is very different from 20/100, but with areas, this is not the case, as one observes one sample of each mixed colour per jar rather than 12 and 120 samples respectively.

The authors make a sharp distinction between "true statistical inference" and "heuristics", but this distinction is not clear to me in the present context. The birds must do the job somehow, namely they must have a perceptive/psychological mechanism to translate what they see to a behavioural choice. They might, for instance, acquire an inhibitory trend towards orange and an appetitive one towards black. The balance between these two tendencies can determine choice probability. Is this a heuristic or a true inference? It seems that their dismissal of heuristics is in fact a dismissal of any realistic algorithm. As a minimum, I'd like to see a more conclusive differentiation between 'true inferential' and 'heuristic' decision making.

A couple of points about the treatment of taxa: When I read the following lines "kea clearly mirrored the type of statistical inferences made by humans and great apes when solving these problems, rather than using heuristics, as monkeys do." and "evidence that capuchins and macaques are not capable of true statistical inference" I was startled by the strength of confidence on what all monkeys are supposed to not being able to do. This is based on a few papers in a couple of monkey species. Affirming the inability of any taxon to do something has a history of failing and warrants caution. People used to say that humans were the only species using tools, and other statements that sound ludicrous today given advanced knowledge. Is it necessary to restate that absence of evidence is not evidence of absence? Comparisons between the cognitive abilities of different species is notoriously difficult and as a minimum requires testing a battery of different protocols, since one species may use a simple algorithm in one context and a complex one (true inference?) in another. Referring to a diverse taxon requires exploring many species within the taxon and with many protocols. Such sweeping statements can be misinterpreted as a bias towards overselling the 'intelligence' of one's working species, always a bad habit.

I was similarly surprised by reading the statement "Much work in this field over the past 15 years has focused mainly on corvids, which have produced their most impressive problem-solving performances predominantly on domain-specific, ecologically relevant tasks, such as those involving caching or tool use^{40, 41}." In the period referred to there has been a huge amount of work on parrots, including the seminal work on African greys by Pepperberg, as well as work comparing kea (the topic of this study)

with corvids on tool using and non-tool using protocols, work on problem solving, social learning and tool use in Goffin cockatoos, work with token exchange in other parrot species, etc etc. This seems a very biased way to place the present study in context.

Alex Kacelnik

Reviewer #3:

Remarks to the Author:

This paper address an interesting question in comparative cognition about the ability of animals to process probabilities and make subsequent inference about the likelihood of an event to happen. The results presented strongly suggest (but see my concerns below) that a parrot species, the kea, not only can compute probabilities but also combine physical and social information to improve their inference. The fact that this species shares this ability with some primates results certainly from an independent evolutionary history that raised the intriguing question of the ecological factors favouring such complex cognition. The second conclusion of the paper is that keas seem to rely on true probabilities rather than absolute counting abilities (heuristic mechanisms) that have not been demonstrated so far in other animals than great apes. Further comparative studies need to be conducted to get a better view of the animal species capable of probabilities estimation as their number may be far larger than thought. This is consequently a paper of high significance for the field. I nevertheless regret that the study did not put absolute numbers and relative frequencies in conflict (instead of equating one) to further understand the mechanisms followed by the keas: do they naturally favour relative or absolute numbers? Here, the results show that they are able to use relative frequencies if necessary to solve a task but not that this is their favoured method as seems to be the case in Humans. Additionally, the number of tokens used are quite large thus making harder to process absolute numbers at the benefice of relative proportions. Why not using lower numbers? A control experiment demonstrating their ability to discriminate and compare such numbers would be ideal but at the minimum, information about the current knowledge on keas' numerical abilities should be provided to convince about their faculty. From a practical point of view, the images provided suggest that, in case of high numbers, many tokens would be hidden by the others which makes an absolute counting harder. I suggest to provide real photographs of the jars if this was not the case. Minor suggestions:

1. It should be stated in the results section of experiment 1 that the initial performances are recorded over the 20 first tests.

2. I do not understand why 63 and 58 tokens were used in the condition 3 of the first experiment instead of the exact same number. Could you explain? In addition, given the number chosen (3 rewarding token for 57 unrewarded), the rewarded tokens were probably hardly perceivable, thus making the conclusion of the whole experiment less convincing.

Table 1: Could you add below the pictures the proportions and exact numbers of each category of tokens ?

Response to Reviewers

We thank our three reviewers for their valuable comments on our manuscript. In the following we address their concerns point-by-point.

Reviewer #1 (Remarks to the Author):

The topic of the present study is highly relevant, up to date, and interesting for a broad readership. So far, the study of intuitive statistics has mainly focused on (human and nonhuman) primates. However, to explore the evolutionary roots of this fundamental capacity, it is important to consider the possibility of convergent evolution and investigate this capacity in a wider range of species. Birds with complex cognitive abilities, such as parrots and corvids, seem promising candidates for finding statistical abilities outside the primate lineage. The present study, therefore, fills an important gap in the literature and will most likely stimulate further research in this area. The paper is well written, clear and concise. I particularly appreciate the authors' effort to test not only for the "standard" capacity of reasoning from population to sample, but also to investigate whether birds can use information from different cognitive domains when drawing statistical inference. Hence, I believe that this is a very promising, comprehensive, and relevant study, and the authors have the data to support most of their claims.

We thank the reviewer for their positive comments on our study.

Introduction: I think the introduction (especially the first paragraph) would benefit from more details. Some terms and concepts are not sufficiently explained for the readership of a multidisciplinary journal such as Nature communications. For instance, the authors never define what they mean with the term "cognitive domain", and many readers may not be familiar with "g". Considering the importance of these terms for the present study, I believe it would be wise to add more explanation in the beginning of the manuscript.

We have now clarified both terms in the introduction, in the following phrases:

Lines 23-26: "There is currently great debate on the extent to which both human and nonhuman intelligence is domain-specific or domain-general: that is, whether subunits of the mind have evolved to solve specific adaptive problems, or whether intelligence evolves more generally, with the same cognitive mechanisms applied flexibly to multiple problems (Cosmides and Tooby, 1987)."

Lines 26-28: "In humans, one source of evidence for domain-general intelligence, rather than domain-specific intelligence, are correlations between performance at different tasks ('g') (Spearman, 1904; Carroll, 1993). Further evidence for domain-generality..."

Page 3, line 37: Teglas et al., 2011 (one of the most convincing demonstrations of infants integrating physical (spatio-temporal) and statistical information).

We apologise for this omission, we did cite this paper in an earlier draft and have now cited it in lines 41-42: "Second, infants can integrate information about physical constraints into their statistical inferences", and in lines 219-221 of our discussion:

“Experiment 2 shows that kea, like infants, can integrate knowledge about a physical barrier into their predictions of a sampling outcome, even though the overall population distributions were identical.”

Page 4, line 48: Eckert, Call, et al., 2018 (the first study to rule out absolute quantity heuristics both regarding preferred and non-preferred food items in great apes); Placi et al., 2019 (long-tailed macaques use relative, rather than absolute frequencies in a statistical reasoning task with a different task format than in previous studies).

We thank the reviewer for bringing these studies to our attention. We now cite the Eckert et al. 2018 study in line 55. The study by Placi and colleagues (2019) shows that long-tailed macaques do not use the absolute number of positive events to make statistical inferences, but fails to directly test whether they are simply avoiding the object associated with the greatest absolute number of negative events. We now cite this accordingly in lines 56-60: “In contrast, capuchins use quantity heuristics based on the absolute frequency of negative items, and it is not yet clear whether rhesus monkeys, long-tailed macaques, pigeons, and African grey parrots use relative frequency or the absolute number of either positive or negative items (or events) when predicting sampling outcomes.”

Page 4, line 51: Roberts et al., 2018 (Pigeons use relative, rather than absolute quantities in a probabilistic reasoning task).

Both Roberts et al. (2018) and Placi et al. (2019) suggest that not only great apes may be capable of true statistical inference -> the authors might want to rephrase the sentence on page 4, lines 47f.

We thank the reviewer for pointing out both these studies. As noted above, the Placi et al. (2019) study does not rule out subjects using the absolute numbers of negative events, while in Roberts et al. (2018) in the crucial control comparing relative and absolute rewards (Experiment 2), pigeons performed no differently from chance throughout their first 54 trials (session 1), with performance only improving to significantly above chance in the following sessions of trials. This is problematic as side keys were always presented in the same location so subjects may have simply learned the rule “peck left” or “peck right” over a large number of trials.

As quoted above, we now cite both these studies in lines 56-60.

Experiment 1: My main criticism is that this study did not test a spontaneous and intuitive ability, but rather trained subjects to reach criteria in the different tasks, which is a crucial difference to previous studies on human infants, great apes, and monkeys. Hence, we do not know whether birds truly inferred the outcome of the sampling event, like primates did, but rather learned an association between one of the populations and a reward over the course of trials. In contrast to ape and monkey studies, the authors do not report first trial performance or test for an effect of trial number on the subjects’ performance in order to shed more light on this possibility.

The reviewer suggests that our results may be due to the kea learning “an association between one of the populations and a reward over the course of trials”. To test this as the

reviewer suggested, we have now added analyses of the effect of trial number, and performance on the first trial to test this hypothesis, to our manuscript's results section.

An associative account should predict performance at chance, i.e. 50%, in the first trial of each condition. We find that kea make the correct choice in their first trial of any given condition in 72.22% of cases. Including only kea that performed above chance over the first 20 trials in that condition, this increases to 81.48% of first trials. We have provided these percentages in our manuscript, in lines 196-199: "Across all conditions and all experiments, subjects' first trial was correct in 72.22% of trials. Taking into account only the subjects that succeeded within the first 20 trials of each condition, first trial performance was correct in 81.48% of cases."

We use a Bayesian model fitted to first trial performance data for all subjects to show that an average kea, given the same training and experience, would perform above chance in their first trial of any given condition. This analysis has now been added to our manuscript: "We fitted an intercept-only Bayesian model to our first trial data for all subjects. When compared against a 0.5 baseline probability of success, our model revealed that the median posterior probability of a randomly-selected kea succeeding within their first trial, regardless of condition, was 0.70 (pMCMC = 0.005)." (Lines 198-201)

An associative account would also predict a positive correlation between trial number and average performance in that trial, across subjects. As in Placì et al. 2019, we correlated average performance in each trial with trial number, within each of our six test conditions. We did not find evidence for a positive correlation for any of our conditions ($BF < 3$), and we found strong support for the null hypothesis of no improvement in performance over the course of the first 20 trials for Experiment 1's Condition 1 ($BF < 0.33$). Furthermore, an associative account would also predict an effect of learning across conditions, leading to a positive correlation between performance and condition order. A correlation between average performance within the first 20 trials of each condition and condition order did not reveal learning effects over the course of the three experiments ($BF = 0.508$).

These analyses have now been added to our manuscript: "We ran Bayesian correlation tests of average performance across the first 20 trials of each condition to examine whether performance increased over the course of the first 20 trials. We found no evidence for learning effects ($BF < 3$; results for each condition's analysis are reported in Supplementary Table 5). Similarly, we examined if performance increased across conditions, but found no correlation between condition number and average performance within the first 20 trials of each condition ($BF = 0.508$). These results show that subjects' performance did not improve over time within or between test conditions, and therefore kea were not relying on low-level associative strategies to make their sampling predictions." (Lines 204-212)

This problem is aggravated by the fact that the authors did not counterbalance the order of conditions- hence, learning across conditions is possible, and perhaps even likely, especially considering that some of the populations remained the same across several conditions. For instance, both in Condition 1 and 2, the unfavorable population contained a 20:100 mixture. After having received on average 120 trials in Condition 1, kea most likely knew that the

20:100 population would not lead to a rewarding token as a sample when it was presented to them again in Condition 2- they did not need to draw any statistical inference. This learning explanation is supported by the fact that birds became gradually better over the course of the three test conditions, a fact that would otherwise be surprising considering that task difficulty increased and the magnitude of difference between the two populations decreased (which would make statistical reasoning presumably more difficult). I understand that for birds the general procedure might be less intuitive/familiar than for primates, and therefore more training might be necessary. However, ideally this training should have been administered as a pre-training only, and not involve the same populations as in the actual test conditions.

As we mentioned above, we provide clear evidence that learning across conditions did not occur. There is also clear evidence that kea did not use the suggested heuristic to make their judgements: if kea had simply learned to avoid a 20:100 rewarding-to-unrewarding population of tokens, we should expect them to perform at chance in all subsequent conditions (Experiment 1's Condition 3, Experiment 2's Condition 1 & 2, and Experiment 3), where there was no such population. We did not see this pattern of performance in our results.

The authors mainly rely on the birds' good results in Condition 2 and 3 when claiming that kea, in contrast to monkeys, are capable of true statistical inference based on relative rather than absolute quantity. However, given the methodological discrepancies between the present study (several hundreds of trials per condition, using the same population across conditions) and primate studies (12 trials per condition, different populations for each condition), I think it is misleading to conclude that kea, but not monkeys, succeeded in statistical inference tasks in which absolute and relative frequencies were disentangled.

We agree it was premature to make such a firm conclusion in regard to the differences between kea and monkey cognition, particularly given that only three monkey species have been studied to date on this paradigm, and so have now reworded our discussion to reflect this: "Birds last shared a common ancestor with humans at least 312 million years ago. This evolutionary distance suggests that domain-general statistical inference may have arisen twice on our planet via convergent evolution. However, further work is required over a wider range of avian and primate species to provide a more accurate evolutionary account of when statistical inference has emerged in different taxa, and so test this hypothesis further." (Lines 265-270)

In regard to the differences in methods, we followed past methodologies in terms of the design of our experiments, and in terms of the same individuals taking part in several conditions (Rakoczy et al. 2014; Tecwyn et al. 2017; Eckert et al. 2018a, 2018b; Placi et al. 2018, 2019). We are not aware of any studies on intuitive statistics in primates where new participants were used for each of the different conditions.

However, the reviewer is correct that past studies with this paradigm have not given subjects further experience at each task until they reached criteria at 17/20 before proceeding to the next condition. This procedure is commonly used during cognitive experiments on birds (e.g. Seed et al. 2006; Teschke et al. 2013; Gruber et al. 2019) as a way to ensure subjects are consistent at a specific condition before being given a novel one. Without such consolidation, it is hard to know if failure at the novel condition is because subjects did not understand it, or because they were simply inconsistent at making the correct choice generally. It is difficult to see how this minor variation in

procedure would have influenced whether kea use statistical inference. This is because this additional experience would only have consolidated whatever strategy the kea were using initially because learning was unstructured. As we stated in our original manuscript: “This is highly intriguing given that the learning of the kea was unstructured: both heuristics and domain-general statistical inference would have worked to solve several of the early problems presented to kea, yet kea clearly mirrored the type of statistical inferences made by humans and the great apes when solving these problems, rather than using heuristics.” That is, using either absolute number or relative frequency would have resulted in above chance performance in Condition 1 of Experiment 1. Giving kea more experience of the condition would only have consolidated the strategy they were using (relative or absolute). The same argument holds for our other conditions.

We now address this in our discussion, where we highlight the differences between the methods here and in past studies in primates, and the implications of these differences (Lines 245-255): “Unlike previous studies in primates, kea were presented with additional training trials before progressing to the next condition. This procedure is commonly used during cognitive experiments on birds (Seed et al. 2006; Teschke et al. 2013; Gruber et al. 2019) to ensure subjects are consistent at a specific condition before being given a novel one. It seems unlikely this additional training would have affected the strategy kea were using to make their sampling predictions across our experiments, as this additional experience would have only consolidated whatever strategy kea were using at the time, be it the use of relative frequencies or quantity heuristics. That is, the learning of the kea was unstructured: both quantity heuristics and domain-general statistical inference would have worked to solve several of the early problems presented to kea, yet kea clearly mirrored the type of statistical inferences made by humans and the great apes when solving these problems.”

I suggest that the authors re-run Condition 2 with new populations (different ratios) to verify their claim. If this is not possible, the discussion should acknowledge the methodological differences between bird and monkey studies, as well as the alternative explanation that birds may have formed associations between populations and rewards in Experiment 1.

As mentioned above, we now acknowledge the differences in methodology between this study and others. Our results cannot be explained by an associative learning strategy as suggested by the reviewer, as we report in our new analyses (Lines 204-212).

Experiment 2: For this experiment I have a question regarding the procedure, in particular what exactly the birds could see: Did they see that the experimenter’s hand reached only into the upper compartment of the jar? Or did they just see that the hand was inserted into the jar, but not into which compartment it reached? I think this question is crucial, because it may tell us how the birds represented the scene. If kea saw that the experimenter’s hand only reached the upper compartment, they did not necessarily have to consider both statistical and physical circumstances in order to solve the task; it would have been sufficient to perceive the two compartments as two separate populations- similar as if they would have been in two different jars. Again, this is an important difference to the infant studies testing for the integration of statistical and physical information (e.g. Teglas et al., 2007; 2011; Denison & Xu 2010; Denison et al., 2014). In these studies, infants inferred the effect of the physical determinant (e.g. barrier) on the sampling event without actually seeing the effect (because the sampling event itself was occluded).

Just as in the infant studies testing for the integration of statistical and physical information (e.g. Denison & Xu 2010; Denison et al., 2014), in our study the sampling event was occluded. Across all experiments, kea only saw the experimenter's hand disappear behind the cardboard occluder (a cardboard lid) on the top of the jar. We now state this explicitly in our methodology section, in lines 327-330: "Throughout training and testing, kea could only see the experimenter's hand disappear behind the cardboard occluder on the top of the jar. Therefore, subjects were unable to see how far down the populations the experimenter's hand reached, or which token it sampled from the population."

If, in the present study, kea did see the effect of the barrier (i.e. that the hand was only inserted in the upper container), the authors should acknowledge this difference to infant studies in their discussion and tune down the conclusions drawn from this experiment. Kea's performance in this experiment is still impressive and a first hint that birds might consider a physical constraint when drawing statistical inference. But the here demonstrated abilities are clearly qualitatively different from those that have been demonstrated in human infants.

Please see our comment above.

Experiment 3: In this experiment I do not understand the rationale of the final experience phase (memory probe) with E2. Why was it beneficial to give kea extra experience with only one of the two experimenters before the actual test?

The beauty of the original experiment (which was developed for chimpanzees; Eckert, Rakoczy, et al. 2018) was that subjects were non-differentially rewarded by the two experimenters in the demonstration phase, before they were allowed to make a choice between them in the test phase. Hence, chimpanzees had no way of associating one of the two experimenters positively or negatively, and instead had to use their statistically acquired knowledge about the experimenters' biases as only source of information.

By contrast, in the present study, the authors included an additional experience phase in between the demonstration phase and the test phase. Crucially, in this phase the birds were given additional experience with only one of the two experimenters (E2), therefore weakening the non-differential character of the demonstration phase. In other words: The latest experience that birds had with each of the experimenters by the time the test started was "E1 always samples rewarding tokens", "E2 sometimes samples rewarding tokens and sometimes unrewarding tokens". Hence, it is unclear whether birds succeeded in the test condition truly by the means of statistically acquired knowledge or rather by a strategy to avoid the experimenter which had given them unrewarding samples in their previous encounter. Again, if the authors are not able to rule out this alternative explanation, they should acknowledge it in the discussion and tune down their interpretations.

We added this memory probe condition between the demonstration and test condition to ensure that kea were still attending to the populations of tokens in the two different jars. Our concern was that, during demonstration trials, kea could have stopped attending to the populations of tokens within the jars and focused entirely on the experimenters' actions. In order to ensure kea had not simply learned to ignore statistical information during the demonstration phase, but rather were integrating it with statistical information, we required them to perform a simple task (comparing an 100% rewarding to an 100% unrewarding population) before the test condition.

However, there was no possibility of the kea learning to “avoid the experimenter which had given them unrewarding samples in their previous encounter”, as the reviewer suggests. The ‘unbiased experimenter’ was used for this memory probe and they would have become more positively associated with food, given all kea performed above chance in their set of 20 probe trials (all subjects scored between 17/20 and 19/20). Thus, if kea had learnt any rule, it would have been ‘approach the experimenter which had given them rewarding samples in their previous encounter’. However, we saw no evidence for such a rule in our data, given that 3/6 kea chose the biased experimenter significantly above chance, and none of our subjects selected the unbiased experimenter significantly above chance.

We have now clarified these points in our methodology section for Experiment 3, in lines 439-441: “Similarly, an increased number of positive “rewarding token” experiences with E2 should make the choice of E1 less likely at test.” and lines 441-444: “This phase ensured that subjects could and would still attend to the contents of jars following the demonstrations, and had not simply learned to ignore jar contents during the demonstration phase.”

Additional remarks: In my opinion, the term integration of information from different cognitive domains is not appropriate. What the authors could show in this study (if the above-mentioned points are sufficiently addressed/clarified) is that kea flexibly decided which of two sources of information (e.g. proportional distribution of items vs. information about a physical barrier) will determine the outcome of a sampling event. An integration of these two sources of information would imply that the influence of each factor is weighted and included in the final computation, leading to a gradual response (similar as in, e.g. Teglas et al., 2011).

The term ‘integration’ is used in the intuitive statistics literature to denote an ability to combine information from the social or physical domains with statistical information, so that a prediction based purely on probabilistic knowledge is overridden in preference of one that also uses this knowledge (Xu & Denison, 2009; Denison & Xu, 2010; Denison et al., 2014). For example, in Denison & Xu 2010 (our underlining), ‘integration’ is used to denote overriding of one domain (statistical inference) by another (social cognition): “These findings suggest that infants were able to integrate domain-specific knowledge regarding agents into a statistical inference mechanism in a meaningful way. They were able to override probabilistic information in favor of domain-specific knowledge...”

In our manuscript, we use the term ‘integration’ in the same way as several studies in the infant literature (e.g. Xu & Denison, 2009; Denison & Xu, 2010; Denison et al., 2014). We now clarify this in our manuscript within our description of the physical and social integration signatures:

“Second, infants can integrate information about physical constraints into their statistical inferences. For example, infants override predictions based purely on relative probabilities when some objects in a population cannot be sampled because they are held back by a physical barrier. Third, infants integrate social information about the preferences of a sampler into their statistical inferences, using their knowledge of an individual’s bias to again override predictions based purely on relative probabilities.” (Lines 41-47)

We also have clarified the following sentence in our discussion: “Therefore, just like infants and the great apes, kea made statistical inferences using relative rather than absolute quantities, and then integrated social and physical information into their predictions, using their knowledge of physical barriers and the bias of a sampler, to override predictions based purely on relative probabilities.” (Lines 226-229)

The reference list contains some formatting errors.

We thank the reviewer for pointing this out – we have now corrected our reference list.

Reviewer #2 (Remarks to the Author):

I enjoyed the paper and the experiments, but had one conceptual difficulty. All the conditions and conclusions are presented and discussed using frequentist descriptions (for instance, in one condition a jar contained 63 +ve and 57 -ve tokens while the other jar the reverse numbers). In fact, it is not possible for any observer (even a human) to ‘count’ the tokens in a jar, since they are distributed randomly in a volume so that some are behind. Further, it is unreasonable to suggest without evidence that the birds count such high numbers. It is more likely that they use the ‘blackness’ and ‘orangeness’ that they can see to categorize the jars. That would be a useful heuristic for the probability of extracting a black or orange bar. In that case, the conditions can be described in terms of relative (or absolute) areas of a given colour. For instance, a jar with 20 +ve and 100 -ve tokens should be described as averaging 16.7% black area. This heuristic is perfectly correlated with changes in both absolute and relative numbers, and leads to using a relative assessment of probability of reward.

I am not clear on whether this change in the description would harm the interpretations the authors put forward, but would like to see that discussed. With frequencies, 2/10 is very different from 20 /100, but with areas, this is not the case, as one observes one sample of each mixed colour per jar rather than 12 and 120 samples respectively.

We thank the reviewer for their positive comments.

As in previous work in intuitive statistics with infants (Xu & Denison, 2009; Gweon et al. 2010; Kushnir et al. 2010; Ma & Xu 2011; Denison et al. 2012; Denison & Xu 2010, 2014; Wellman, 2016) and primates (Rakoczy et al. 2014; Tecwyn et al. 2017; Eckert et al. 2018a, 2018b; Placi et al. 2018, 2019; De Petrillo et al. 2019), we used large sets of numbers as these studies test for the ability of subjects to use frequencies, rather than absolute numbers, to make statistical inferences,

It is currently unclear how infants, apes and the kea in this study extract statistical information. In particular, as Denison and Xu (2014) note, it is not yet clear if subjects make inferences using discrete or continuous quantity representations, but this does not detract from the results here: either way, subjects were extracting information about the relative frequencies of objects, and using it to make predictions about the relative probability of reward (Experiment 1), and then integrated these judgements with both physical (Experiment 2) and social (Experiment 3) information. One key area for future work will be determining exactly how statistical information is extracted and

represented by humans, apes and kea. We have now added this text to our discussion (lines 256-264).

As previous studies in this field have used ratios, we now provide ratio information alongside the absolute numbers of rewarding and unrewarding tokens, summarised in Table 1.

The authors make a sharp distinction between “true statistical inference” and “heuristics”, but this distinction is not clear to me in the present context. The birds must do the job somehow, namely they must have a perceptive/psychological mechanism to translate what they see to a behavioural choice. They might, for instance, acquire an inhibitory trend towards orange and an appetitive one towards black. The balance between these two tendencies can determine choice probability. Is this a heuristic or a true inference? It seems that their dismissal of heuristics is in fact a dismissal of any realistic algorithm. As a minimum, I’d like to see a more conclusive differentiation between ‘true inferential’ and ‘heuristic’ decision making.

We apologise for the lack of clarity in specifying what we meant by ‘heuristics’. In this context, we meant that rather than kea using information about the relative frequencies of objects, they were relying on simpler alternatives which do not involve comparisons between ratios. Namely, these are: (1) selecting populations with the greater absolute number of positive items rather than comparing the proportions of items within and between populations, and (2) avoiding populations with the smallest number of negative items rather than comparing the ratios within and between populations. We now clearly specify what we mean by ‘heuristics’ in this context in lines 53-56 of our introduction: “...true statistical inference, as they use the relative numbers of items within and between populations when predicting sampling events, rather than using quantity heuristics based on the absolute number of positive or negative objects.” We now further specify the term as “quantity heuristics” throughout the text, as in Placi et al. 2019.

As in previous literature, we use the term ‘inference’ to mean that subjects made a prediction about the outcome of an event on which they had limited information, based on logic or reasoning. We now clarify this in our introduction, lines 34-36: “Making inferences about uncertainty involves generating logical predictions about future events based on limited information.”

A simpler perceptual strategy, where subjects simply selected the hand coming out from the jar with the greatest ‘appetitive’ value, or avoided the hand coming out from the jar with the greatest ‘inhibitory’ value, would have resulted in chance performance throughout Experiments 2 and 3, where the jars contained both the same absolute number / absolute area coverage, and the same frequency, of rewarding and unrewarding tokens. Importantly, kea integrated both physical barriers and information about a sampler’s bias to flexibly update their predictions, showing that they reasoned about the probabilistic aspects of these tasks. Therefore, a simple perceptual explanation is inconsistent with the results obtained for Experiment 2 and Experiment 3.

We therefore show that kea clearly make logic-based, true statistical inferences, using relative rather than absolute quantities, given that (a) they do not use simpler quantity heuristics such as avoiding the largest quantity / area of unrewarding items or selecting

the largest quantity / area of rewarding items to make such judgements, and (b) they flexibly integrate physical and social information into their sampling predictions even when the absolute number of rewarding and unrewarding tokens were identical in both number and coverage area.

A couple of points about the treatment of taxa: When I read the following lines “kea clearly mirrored the type of statistical inferences made by humans and great apes when solving these problems, rather than using heuristics, as monkeys do.” and “evidence that capuchins and macaques are not capable of true statistical inference” I was startled by the strength of confidence on what all monkeys are supposed to not being able to do. This is based on a few papers in a couple of monkey species. Affirming the inability of any taxon to do something has a history of failing and warrants caution. People used to say that humans were the only species using tools, and other statements that sound ludicrous today given advanced knowledge. Is it necessary to restate that absence of evidence is not evidence of absence? Comparisons between the cognitive abilities of different species is notoriously difficult and as a minimum requires testing a battery of different protocols, since one species may use a simple algorithm in one context and a complex one (true inference?) in another. Referring to a diverse taxon requires exploring many species within the taxon and with many protocols. Such sweeping statements can be misinterpreted as a bias towards overselling the ‘intelligence’ of one’s working species, always a bad habit.

We agree with the reviewer that the phrasing was prematurely decisive. We agree that lack of evidence does not provide evidence of absence of an ability in any species, and that context modulates much of animals’ performance in cognitive tasks. We have amended the two sentences mentioned to reflect this:

Lines 254-255: “...kea clearly mirrored the type of statistical inferences made by humans and the great apes when solving these problems.”

We also clarify in our discussion that further work on other primate and bird species will provide a clearer evolutionary pattern for this ability: “Birds last shared a common ancestor with humans at least 312 million years ago. This evolutionary distance suggests that domain-general statistical inference may have arisen twice on our planet via convergent evolution. However, further work is required over a wider range of avian and primate species to provide a more accurate evolutionary account of when statistical inference has emerged in different taxa, and so test this hypothesis further.” (Lines 265-270)

I was similarly surprised by reading the statement “Much work in this field over the past 15 years has focused mainly on corvids, which have produced their most impressive problem-solving performances predominantly on domain-specific, ecologically relevant tasks, such as those involving caching or tool use^{40, 41}.” In the period referred to there has been a huge amount of work on parrots, including the seminal work on African greys by Pepperberg, as well as work comparing kea (the topic of this study) with corvids on tool using and non-tool using protocols, work on problem solving, social learning and tool use in Goffin cockatoos, work with token exchange in other parrot species, etc etc. This seems a very biased way to place the present study in context.

We apologise for this, as it was not our intention to misrepresent the literature. We did discuss parrot cognition in detail in an earlier draft, but removed it due to space constraints. We now cite the studies mentioned by the reviewer in lines 231-240: “Much work in this field over the past 15 years has focused on corvids, which, as a group, have produced their most impressive problem-solving performances predominantly on domain-specific, ecologically relevant tasks, such as those involving caching or tool use (Dickinson & Clayton, 1998; Emery & Clayton, 2001; Chappell & Kacelnik, 2002; Dally et al., 2006; Weir & Kacelnik, 2006; Taylor et al., 2007; Alexis et al. 2007; Wimpenny et al., 2009; von Bayern et al., 2018). Parrots have only recently become the focus of a sustained, global research effort (Auersperg & von Bayern, 2019). Behaviour suggestive of more domain-general processes have emerged in both groups from studies examining the ability of non-tool users to solve tool problems (Bird et al. 2009; Auersperg et al. 2010; Auersperg et al. 2011a, 2011b; Auersperg et al. 2012; Auersperg et al. 2016; Laumer et al. 2016). Our results both support these claims and greatly extend them, in showing that the integration of very different types of information – concerning physical barriers and social preferences – into statistical inferences is possible in at least the mind of one parrot species.”

Reviewer #3 (Remarks to the Author):

This paper address an interesting question in comparative cognition about the ability of animals to process probabilities and make subsequent inference about the likelihood of an event to happen. The results presented strongly suggest (but see my concerns below) that a parrot species, the kea, not only can compute probabilities but also combine physical and social information to improve their inference. The fact that this species shares this ability with some primates results certainly from an independent evolutionary history that raised the intriguing question of the ecological factors favouring such complex cognition. The second conclusion of the paper is that keas seem to rely on true probabilities rather than absolute counting abilities (heuristic mechanisms) that have not been demonstrated so far in other animals than great apes. Further comparative studies need to be conducted to get a better view of the animal species capable of probabilities estimation as their number may be far larger than thought. This is consequently a paper of high significance for the field.

We thank the reviewer for their positive comments.

I nevertheless regret that the study did not put absolute numbers and relative frequencies in conflict (instead of equating one) to further understand the mechanisms followed by the keas: do they naturally favour relative or absolute numbers? Here, the results show that they are able to use relative frequencies if necessary to solve a task but not that this is their favoured method as seems to be the case in Humans.

We hope we are not misunderstanding the reviewer’s comment, but in our study, we did put absolute number and relative frequency into direct conflict in several conditions.

In Experiment 1’s Condition 1, both relative and absolute strategies would have resulted in the correct prediction (selecting the jar with 100 rewarding tokens and 20 unrewarding tokens). Kea were given this task until they reached 17/20 performance, reinforcing whichever method they preferred (relative or absolute) over several trials. Condition 2 then directly tested whether the rule kea had learnt from the previous

condition was ‘select the population with the greatest number of rewarding tokens’. If they had, this would lead to performance at chance, rather than above chance, in this condition. Therefore, Condition 2 directly pitted an absolute number strategy and a relative frequency strategy against each other. Our results provide clear evidence that the kea had naturally favoured using a relative frequency strategy to solve a probabilistic task, given that both were equally useful in Condition 1. We now clarify this in our discussion: “That is, the learning of the kea was unstructured: both quantity heuristics and domain-general statistical inference would have worked to solve several of the early problems presented to kea, yet kea clearly mirrored the type of statistical inferences made by humans and the great apes when solving these problems.” (Lines 252-255)

Condition 3 then tested a relative frequency strategy against the opposite absolute number strategy, ‘avoid the jar with the greatest number of unrewarding tokens’. This strategy would have led to performance at chance in this condition, given that the number of unrewarding tokens in the two populations was identical. Note that this strategy would have been equally successful as using relative frequencies in both Conditions 1 and 2. Therefore, again we directly pitted a relative and absolute strategy, and found clear evidence that kea had preferred to use the relative frequency strategy in Conditions 1 and 2, rather than an absolute number strategy.

These results are further strengthened by Experiments 2 and 3, where the absolute number of tokens in both jars within each condition were always identical. Kea relied on a strategy other than comparing the absolute number of tokens to solve these conditions within their first 20 trials: namely, integrating physical and social information into the perceived relative frequencies of rewarding and unrewarding tokens.

Additionally, the number of tokens used are quite large thus making harder to process absolute numbers at the benefice of relative proportions. Why not using lower numbers? A control experiment demonstrating their ability to discriminate and compare such numbers would be ideal but at the minimum, information about the current knowledge on keas’ numerical abilities should be provided to convince about their faculty.

As in previous work in intuitive statistics with infants (Xu & Denison, 2009; Gweon et al. 2010; Kushnir et al. 2010; Ma & Xu 2011; Denison et al. 2012; Denison & Xu 2010, 2014; Wellman, 2016) and primates (Rakoczy et al. 2014; Tecwyn et al. 2017; Eckert et al. 2018a, 2018b; Placi et al. 2018, 2019; De Petrillo et al. 2019), we used large sets of numbers in our study. In order to make accurate predictions for the sampling events they observed, subjects in these studies must rely on the frequencies of the rewarding and unrewarding objects.

We now clarify that reasoning under uncertainty usually occurs when encountering a large number of objects in lines 38-41: “First, when observing sampling events with a large number of objects, infants show true statistical inference, using the relative frequency of objects in a population to infer the most likely sampling outcome, rather than using quantity heuristics based on the absolute number of objects.”

We make no claims about the numerical cognition of the kea, as our study focused on their ability to extract statistical information from the environment. We now state in

our manuscript: “It is currently unclear how infants, apes, and the kea in this study extract statistical information. In particular, as Denison and Xu note, it is not yet clear if subjects make inferences using discrete or continuous quantity representations, but this does not detract from the results here: either way, subjects were extracting information about the relative frequencies of objects, using it to make predictions about the relative probability of reward (Experiment 1), and then integrating these judgements with both physical (Experiment 2) and social (Experiment 3) information. One key area for future work will be determining exactly how statistical information is extracted and represented by humans, apes and kea.” (Lines 256-264).

From a practical point of view, the images provided suggest that, in case of high numbers, many tokens would be hidden by the others which makes an absolute counting harder. I suggest to provide real photographs of the jars if this was not the case.

As in previous studies of probabilistic inference in infants (Xu & Denison, 2009, Gweon et al. 2010, Kushnir et al. 2010, Ma & Xu 2011, Denison et al. 2012, Denison & Xu 2010, 2014, Wellman, 2016) and non-human animals (Rakoczy et al. 2014, Tecwyn et al. 2017, Eckert et al. 2018a, 2018b, Placi et al. 2018, Placi et al. 2019, De Petrillo et al. 2019), tokens were partially or fully hidden by other tokens within containers. In these studies, subjects were expected to use the proportions of objects present rather than the absolute number of objects to make their sampling predictions, and therefore were not required to count the tokens individually. If all tokens had been fully visible and of a small enough number to count, it would have been impossible to sample a single item from a container without the subject knowing exactly which item it was – the sampling would have to occur in full view – and it would be impossible to rule out the possibility that the animals might be using the absolute number of items rather than relative frequencies to solve these problem. In our study, as in the previous work in primates and infants, the sampling event was always occluded in all training and test trials, and kea always saw a representative sample of the population for each trial, with at least two of the minority tokens fully visible in all trials.

We have now made this clear in our manuscript: “Throughout training and testing, kea could only see the experimenter’s hand disappear behind the cardboard occluder on the top of the jar. Therefore, subjects were unable to see how far down the populations the experimenter’s hand reached, or which token it sampled from the population. In test conditions with very disparate ratios of rewarding to unrewarding tokens, we ensured that at least two tokens from the minority population were fully visible to the subjects in every trial.” (Lines 327-332)

Minor suggestions:

1. *It should be stated in the results section of experiment 1 that the initial performances are recorded over the 20 first tests.*

We thank the reviewer for pointing out this omission – we now clearly state within the results section of experiment 1 that only the first 20 trials of each condition were analysed, on lines 81-82: “Three of six kea spontaneously showed a preference for the hand that had sampled the population with 100 rewarding tokens within their first 20 trials.”

2. *I do not understand why 63 and 58 tokens were used in the condition 3 of the first experiment instead of the exact same number. Could you explain? In addition, given the number chosen (3 rewarding token for 57 unrewarded), the rewarded tokens were probably hardly perceivable, thus making the conclusion of the whole experiment less convincing.*

We utilised new proportions and absolute numbers of tokens in Condition 3 of Experiment 1 to ensure that kea had not simply learned to avoid a particular population in previous conditions. At least two minority tokens were always fully visible to the subjects in every trial, regardless of the population ratio. Therefore, even when the population of 57 unrewarding to 3 rewarding tokens was presented, subjects could still clearly see that rewarding tokens continued to exist in this population. We thank the reviewer for pointing out that we had not previously clarified this in our manuscript – we now state this point in our methodology section, in lines 330-332: “In test conditions with very disparate ratios of rewarding to unrewarding tokens, we ensured that at least two tokens from the minority population were fully visible to the subjects in every trial.”

3. *Table 1: Could you add below the pictures the proportions and exact numbers of each category of tokens?*

We agree with the reviewer that this will greatly improve the clarity of our manuscript. We have now added both the population ratios and absolute numbers of tokens under the illustrations for each condition in Table 1.

Reviewers' Comments:

Reviewer #1:

Remarks to the Author:

The authors have addressed most of my previously expressed concerns and I agree with the overall conclusion that kea have demonstrated a capacity for statistical inference and that kea are able to consider information from different domains when drawing statistical inferences. However, I still have some issues with the terminology used by the authors (see below).

My main criticism was that the design of the study differed from previous studies testing human infants and non-human primates in various ways. Most importantly, I was concerned that the large number of trials within each condition, the "training until reaching criterion", the (not counterbalanced) order of conditions, and the fact that not all conditions used novel populations of tokens might have facilitated associative learning and, therefore, that keas' performance might not reflect true statistical abilities.

The authors have now added information about first trial performance to the manuscript. This information shows that kea did in fact prefer the "correct" sample from the first trial onwards in all conditions. Moreover, the authors conducted additional analyses showing that subjects' performance did not improve over trials within conditions, neither did it improve across conditions. Hence, it seems unlikely that the birds used associative learning mechanisms to solve the tasks.

In my last revision I wrote: "However, given the methodological discrepancies between the present study (several hundreds of trials per condition, using the same population across conditions) and primate studies (12 trials per condition, different populations for each condition), I think it is misleading to conclude that kea, but not monkeys, succeeded in statistical inference tasks in which absolute and relative frequencies were disentangled".

The authors responded by stating that: "In regard to the differences in methods, we followed past methodologies in terms of the design of our experiments, and in terms of the same individuals taking part in several conditions (Rakoczy et al. 2014; Tecwyn et al. 2017; Eckert et al. 2018a, 2018b; Placi et al. 2018, 2019). We are not aware of any studies on intuitive statistics in primates where new participants were used for each of the different conditions."

There seems to be a misunderstanding here. With "different populations" I referred to the populations of tokens used across conditions, and emphasized that the authors risked that their birds would solve the tasks by means of associative learning when using the same population in several conditions. The authors apparently thought that I was criticizing them for testing the same population of kea across experiments. This was not my intention. I am aware of the fact that it is nearly impossible to have enough testable individuals in order to conduct a similar study using a between-subjects design for most species. I also don't think that this is necessary, as long as associative learning is sufficiently controlled for.

I am satisfied with the new first trial data/analysis and to me this sufficiently addresses my associative learning concern.

I also appreciate that the authors tuned down their conclusions regarding differences between monkeys and kea and now acknowledge the discrepancies in study design in the discussion.

In my last revision I wrote that, in my opinion, what keas do in this study is not integrating different sources of information with statistical inference, but rather flexibly deciding which of two sources of information will determine the outcome of the event. The authors responded by stating that the term 'integration' is commonly used in the intuitive statistics literature in similar studies. To support their

claim the authors cite three papers by Denison and Xu: Xu & Denison, 2009; Denison & Xu, 2010; Denison et al., 2014.

However, the very same authors state in a review with regard to both the barrier experiment by Teglas et al., 2007 (after which the current Experiment 2 was designed) and the experimenter bias experiment by Xu and Denison, 2009 (after which the current Experiment 3 was designed):

“In Teglas et al. (2007) and Xu and Denison (2009), infants were able to assess when to use ratio information and when to use domain-specific knowledge regarding naïve physics or naïve psychology to make an inference. These abilities are impressive as infants used very subtle cues- the presence or absence of a physical barrier and an agent’s perceptual access to a scene- to infer whether or not they should compute probabilities or instead make a judgement using physical or psychological knowledge, respectively. However, these experiments did not require infants to fully integrate domain-specific knowledge in statistical inference- that is, infants were required to use one knowledge source or the other but not both simultaneously.” (Denison & Xu, 2012, p.41).

In order to convincingly demonstrate that infants truly integrate, e.g. statistical and physical information, different task setups were used (see Denison et al., 2014; Denison & Xu, 2010; Teglas et al., 2011).

Therefore, I am still not convinced that the current study shows that kea can integrate statistical and other types of information. This does not devalue the results; I think it is extremely impressive that kea a) are capable of statistical inference and b) can flexibly decide which of different factors is the important one regarding the outcome of an event.

Another terminology issue arises from the word “preference” which the experimenters use with regard to the experimenters’ choice biases in Experiment 3. Attributing preferences or desires to others does imply “theory of mind”-like abilities, something that has not been addressed in this study. We cannot disentangle whether the birds attributed a preference for one type of token to the biased experimenter, or whether they merely noticed that this experimenter, for whatever reason, always sampled one type of token.

Previous work on infants and chimpanzees has administered additional experiments in which subjects adjusted their choice when dealing with a biased experimenter depending on whether this experimenter had visual access to the populations or sampled blindly. Such an experiment is missing in the current study, therefore we cannot draw any conclusions about an attribution of mental states by kea.

Hence, I think it would be more appropriate to use the term “bias” instead of “preference” throughout the manuscript.

In line 243 the authors write about “social preferences”. This term usually refers to differences in degrees of altruism, fairness, reciprocity, etc. I believe that the authors mean “preferences of others” (which, again, should more accurately be “others’ choice biases”).

Reviewer #2:

Remarks to the Author:

The authors have clarified many parts of the text and solved most of the difficulties in the earlier version.

I still have differences with (or perhaps just don’t understand) their interpretation of what they mean by “quantity heuristics” and “making inferences about uncertainty [that] involves generating logical predictions about future events based on limited information”.

Suppose that, using conventional psychological terminology, a subject faces a mixture of S+ and S- stimuli, encountered sequentially in a random order, in a given proportion. Say that the subject keeps a running average with a recency discounting process, as in the ‘linear operator’ literature on choice

derived last century from the Bush and Mosteller algorithms. In that case, the estimate that drives preference is a time-weighted function of the ratio of the two kinds of stimuli, not of only the absolute numbers of either S+ or S-. Would that be called by the authors a quantitative heuristic or true inference? Would they still claim that this is a rare phenomenon only known in primates? Such a mechanism would make BF Skinner happy and work for pigeons, rats, etc. It has been tested with different variants in many species. All studies examining animals' response to probability speak to this concept. I fail to see the critical difference with 'true logical inference'.

It could be argued that in the present experiments the stimuli were encountered simultaneously rather than sequentially as in the Bush and Mosteller linear operator literature, but since the subjects somehow "counted" the coloured balls, they must have somehow treated the two kinds of stimuli as contributing to an estimate. Here the physical presentation of the balls in jars and the large number of items, made impossible for the subjects to see all balls present.

It would appear that what the authors call a 'simple quantity heuristic' would be for the subject to neglect either S+ or S- in deriving a moving average, as this would then not represent the ratio but just absolute numbers of one or the other. But I see no attempt to examine how and whether the kea are counting.

In summary, I try, but fail, to understand why being sensitive to a property of the mixture (say the 'orangeness' they see), which is a function of the ratio, is more surprising or more sophisticated than counting the absolute numbers, which they would downgrade to being a heuristic.

I am totally happy to let my comment stand as a difficulty of this particular referee in understanding the interpretation of 'true inference' rather than a definite failure of the study. The results do stand, it is the theoretical framing and the conclusions that I have difficulties with.

Alex Kacelnik

Reviewer #3:

Remarks to the Author:

The authors have addressed my comments satisfactorily.

Response to Reviewers

We thank our three referees for their second round of reviews and helpful comments. In the following we address their concerns point-by-point.

Reviewer #1 (Remarks to the Author):

The authors have addressed most of my previously expressed concerns and I agree with the overall conclusion that kea have demonstrated a capacity for statistical inference and that kea are able to consider information from different domains when drawing statistical inferences. However, I still have some issues with the terminology used by the authors (see below).

My main criticism was that the design of the study differed from previous studies testing human infants and non-human primates in various ways. Most importantly, I was concerned that the large number of trials within each condition, the “training until reaching criterion”, the (not counterbalanced) order of conditions, and the fact that not all conditions used novel populations of tokens might have facilitated associative learning and, therefore, that keas’ performance might not reflect true statistical abilities.

The authors have now added information about first trial performance to the manuscript. This information shows that kea did in fact prefer the “correct” sample from the first trial onwards in all conditions. Moreover, the authors conducted additional analyses showing that subjects’ performance did not improve over trials within conditions, neither did it improve across conditions. Hence, it seems unlikely that the birds used associative learning mechanisms to solve the tasks.

We thank the reviewer for their positive comments.

In my last revision I wrote: “However, given the methodological discrepancies between the present study (several hundreds of trials per condition, using the same population across conditions) and primate studies (12 trials per condition, different populations for each condition), I think it is misleading to conclude that kea, but not monkeys, succeeded in statistical inference tasks in which absolute and relative frequencies were disentangled”.

The authors responded by stating that: “In regard to the differences in methods, we followed past methodologies in terms of the design of our experiments, and in terms of the same individuals taking part in several conditions (Rakoczy et al. 2014; Tecwyn et al. 2017; Eckert et al. 2018a, 2018b; Placì et al. 2018, 2019). We are not aware of any studies on intuitive statistics in primates where new participants were used for each of the different conditions.” There seems to be a misunderstanding here. With “different populations” I referred to the populations of tokens used across conditions, and emphasized that the authors risked that their birds would solve the tasks by means of associative learning when using the same population in several conditions. The authors apparently thought that I was criticizing them for testing the same population of kea across experiments. This was not my intention. I am aware of the fact that it is nearly impossible to have enough testable individuals in order to conduct a similar study using a between-subjects design for most species. I also don’t think that this is necessary, as long as associative learning is sufficiently controlled for. I am satisfied with the new first trial data/analysis and to me this sufficiently addresses my associative learning concern.

We thank the reviewer for the clarification, as we had misunderstood their wording.

I also appreciate that the authors tuned down their conclusions regarding differences between monkeys and kea and now acknowledge the discrepancies in study design in the discussion.

In my last revision I wrote that, in my opinion, what keas do in this study is not integrating different sources of information with statistical inference, but rather flexibly deciding which of two sources of information will determine the outcome of the event. The authors responded by stating that the term ‘integration’ is commonly used in the intuitive statistics literature in similar studies. To support their claim the authors cite three papers by Denison and Xu: Xu & Denison, 2009; Denison & Xu, 2010; Denison et al., 2014.

However, the very same authors state in a review with regard to both the barrier experiment by Teglas et al., 2007 (after which the current Experiment 2 was designed) and the experimenter bias experiment by Xu and Denison, 2009 (after which the current Experiment 3 was designed):

“In Teglas et al. (2007) and Xu and Denison (2009), infants were able to assess when to use ratio information and when to use domain-specific knowledge regarding naïve physics or naïve psychology to make an inference. These abilities are impressive as infants used very

subtle cues- the presence or absence of a physical barrier and an agent's perceptual access to a scene- to infer whether or not they should compute probabilities or instead make a judgement using physical or psychological knowledge, respectively. However, these experiments did not require infants to fully integrate domain-specific knowledge in statistical inference- that is, infants were required to use one knowledge source or the other but not both simultaneously."

(Denison & Xu, 2012, p.41).

In order to convincingly demonstrate that infants truly integrate, e.g. statistical and physical information, different task setups were used (see Denison et al., 2014; Denison & Xu, 2010; Teglas et al., 2011).

Therefore, I am still not convinced that the current study shows that kea can integrate statistical and other types of information. This does not devalue the results; I think it is extremely impressive that kea a) are capable of statistical inference and b) can flexibly decide which of different factors is the important one regarding the outcome of an event.

We agree with the reviewer that there is some ambiguity in the literature as to what can be considered integration. This ambiguity is clear in the paper the reviewer cites, Denison & Xu (2012), , in that the authors state, when discussing the sampling bias results from Xu & Garcia (2008) where social information was shown to override frequency information, that, “infants in this experiment flexibly took into account sampling conditions and integrated substantive domain knowledge about agents into probabilistic inference” before then going on to say, when discussing Teglas et al. (2007) and Xu & Denison (2009) “these experiments did not require infants to fully integrate domain-specific knowledge in statistical inferenced, that is, infants were required to use one knowledge source or the other but not both simultaneously.” We therefore think it would be useful to make a distinction between two types of integration: one where one source of information overrides the predictions made by another, the second where two knowledge sources are considered simultaneously.

We now discuss this distinction in our discussion: “It is important to note, however, that the level of information integration across our experiments was different. Experiment 2 tested if kea could simultaneously integrate information about relative frequency with the presence of a barrier in order to make a judgement, because relying solely on one source of information – either the presence of a barrier, or the overall frequency of the

population – would have led to the incorrect choice. In Experiment 3, however, kea needed to override their prior reliance on relative frequency information in favour of information about the sampling bias of an experimenter. Our results therefore suggest that kea can fully integrate physical information with relative frequency by using both sources of information simultaneously, and can integrate social information by overriding relative frequency information with social information. However, more work is needed to show if kea can also simultaneously consider social and relative frequency information, or can make judgements that simultaneously combine social, physical and relative frequency information together.” (Lines 244-256)

Another terminology issue arises from the word “preference” which the experimenters use with regard to the experimenters’ choice biases in Experiment 3. Attributing preferences or desires to others does imply “theory of mind”-like abilities, something that has not been addressed in this study. We cannot disentangle whether the birds attributed a preference for one type of token to the biased experimenter, or whether they merely noticed that this experimenter, for whatever reason, always sampled one type of token.

Previous work on infants and chimpanzees has administered additional experiments in which subjects adjusted their choice when dealing with a biased experimenter depending on whether this experimenter had visual access to the populations or sampled blindly. Such an experiment is missing in the current study, therefore we cannot draw any conclusions about an attribution of mental states by kea.

Hence, I think it would be more appropriate to use the term “bias” instead of “preference” throughout the manuscript.

In line 243 the authors write about “social preferences”. This term usually refers to differences in degrees of altruism, fairness, reciprocity, etc. I believe that the authors mean “preferences of others” (which, again, should more accurately be “others’ choice biases”).

We now refer to experimenter’s preferences within our experiments as ‘biases’ or ‘sampling biases’ throughout the manuscript.

Reviewer #2 (Remarks to the Author):

The authors have clarified many parts of the text and solved most of the difficulties in the earlier version.

I still have differences with (or perhaps just don't understand) their interpretation of what they mean by "quantity heuristics" and "making inferences about uncertainty [that] involves generating logical predictions about future events based on limited information".

Suppose that, using conventional psychological terminology, a subject faces a mixture of S+ and S- stimuli, encountered sequentially in a random order, in a given proportion. Say that the subject keeps a running average with a recency discounting process, as in the 'linear operator' literature on choice derived last century from the Bush and Mosteller algorithms. In that case, the estimate that drives preference is a time-weighted function of the ratio of the two kinds of stimuli, not of only the absolute numbers of either S+ or S-. Would that be called by the authors a quantitative heuristic or true inference? Would they still claim that this is a rare phenomenon only known in primates? Such a mechanism would make BF Skinner happy and work for pigeons, rats, etc. It has been tested with different variants in many species. All studies examining animals' response to probability speak to this concept. I fail to see the critical difference with 'true logical inference'.

It could be argued that in the present experiments the stimuli were encountered simultaneously rather than sequentially as in the Bush and Mosteller linear operator literature, but since the subjects somehow "counted" the coloured balls, they must have somehow treated the two kinds of stimuli as contributing to an estimate. Here the physical presentation of the balls in jars and the large number of items, made impossible for the subjects to see all balls present.

It would appear that what the authors call a 'simple quantity heuristic' would be for the subject to neglect either S+ or S- in deriving a moving average, as this would then not represent the ratio but just absolute numbers of one or the other. But I see no attempt to examine how and whether the kea are counting.

In summary, I try, but fail, to understand why being sensitive to a property of the mixture (say the 'orangeness' they see), which is a function of the ratio, is more surprising or more sophisticated than counting the absolute numbers, which they would downgrade to being a heuristic.

I am totally happy to let my comment stand as a difficulty of this particular referee in understanding the interpretation of 'true inference' rather than a definite failure of the study.

The results do stand, it is the theoretical framing and the conclusions that I have difficulties with.

We thank the reviewer for their positive comments and agree with them that the distinction between heuristics and reasoning in this field is a complex one. The quantity heuristic has been suggested to be a simpler way of solving statistical inference problems many times in the literature to date (e.g. Denison & Xu 2014; Girotto et al. 2016; Rakoczy et al. 2014; Tecwyn et al. 2017; Denison & Xu 2019), and it is one that we were able to control for in the design of our experiments, where we have clear evidence kea extract information about relative frequency, rather than absolute number. We think the issues in regard to the quantity heuristic might be arising because the reviewer seems to be assuming that use of a quantity heuristic requires counting, and thus makes the reasonable claim that the use of quantify heuristic might be more complex than the use of frequency information. However, just as it is unclear whether kea (and children or apes) use continuous or discrete information about relative frequency, so it is also unclear whether use of a simpler quantity heuristic would involve the use of discrete information (as via counting) or continuous information. While not the focus of our study, future work examining this possibility in infants, apes, monkeys and birds would be highly useful in clarifying this matter.

Reviewer #3 (Remarks to the Author):

The authors have addressed my comments satisfactorily.

We thank the reviewer for their comments, which have greatly improved the manuscript.

Reviewers' Comments:

Reviewer #1:

Remarks to the Author:

The authors have addressed my concerns satisfactorily. I still think that the term "integration" is not the best fit to describe what the birds are doing, but I am content with the newly added discussion of the terminology issues.